# Investigating and promoting health behaviors reactivity among Hong Kong older adults in the post-COVID-19 Era: An exploratory network analysis

**Ming Yu Claudia Wong**[1], **Kai-ling Ou**[2]*, **Ka Man Leung**[1]

**1** The Education University of Hong Kong, Hong Kong, China, **2** Hong Kong Baptist University, Hong Kong, China

* 21482268@life.hkbu.edu.hk

## Abstract

### Background

Physical distance has increased the risk of developing a variety of health problems, especially among older people. During the COVID-19 lockdown period, physical activity decreased, screen time increased, food consumption increased, as well as exposure to unhealthy behaviour, leading to poorer sleep quality and more negative emotions, which ultimately led to poorer physical health, mental health, and subjective vitality among older adults. Although there were numerous research studies on changes in health behaviours during COVID-19, only a few were targeted at older adults, and none were conducted in Asian countries.

### Method

The current study is aiming to identify the changes in health behaviours, as well as their associations with older adults' mental health as a result of the COVID-19 pandemic among Hong Kong older adults, using exploratory network analysis. In this study, a random tele-survey was conducted among older adults.

### Results

A total of 664 participants have been involved in the telephone survey, with 213 males and 451 females, of which mean age was 74.37. The explored network showed strong centrality and edge stability, thus enabling to reveal an overview of the associations between health behaviours and mental well-being of older adults. The lavaan model has also demonstrated the casual paths within the explored network, which indicated the significant impact of sleeping quality, eating habits and social networking on resilience.

### Conclusion

The outcomes of the study were able to identify the lifestyle changes of Hong Kong older adults due to COVID-19. Although the present study is not regarded as novel internationally,

**Funding:** Funding sources and related paper presentations: Hong Kong Baptist University: Research development fund RC-RDF-RNHA202103 ~ Research Network on Healthy Ageing - Chung Pak Kwong.

**Competing interests:** NO authors have competing interests Enter: The authors have declared that no competing interests exist.

it may be representative of the Hong Kong community. In order to facilitate active aging during the pandemic, a user-friendly electronic platform for healthy living should include older adults in the future.

## Background

Lockdowns and social restrictions during the COVID-19 pandemic may have limited outdoor and social activities, resulting in potential health risks. According to the World Health Organization statement—"Older people are at higher risk from COVID-19" [1], older adults were most likely to avoid being outside for exercise or social activities. Physical distancing could increase the risk of developing a variety of health problems, particularly in older people [2]. In Hong Kong, researchers investigated that the prevalence of sarcopenia in community-dwelling oldest population was high under the physical distancing policy is in force due to lack of physical activity [3].

Other than physical health, social distance also causes negative impacts on the mental health of older adults [4]. A study revealed that the lack of social relatedness during COVID-19 was expected to affect individuals' self-determination and subjective well-being, thus leading to the impact on health behaviours [5]. Recent studies have also identified the changes in people's lifestyle and dietary patterns during the COVID-19 pandemic [6–9]. The researchers commented that nearly half of the respondents reported a decrease in physical activity, increase in screen time as well as food consumption; also, people aged over 40 showed more exposure towards unhealthy behaviours [7]. While, being less physically active, more sedentary behaviour, poorer sleeping quality and diet were associated with more negative mood, hence linking with poorer physical health, mental health and subjective vitality [6,8]. Additionally, during the lockdown period of COVID-19, the negative impacts of loneliness on older adults' mental health status were also identified [10] Notwithstanding the variety of research studies on the changes in health behaviours during COVID-19, studies targeting older adults only were limited and nor in the Asian countries. Therefore, a survey on the changes in health behaviours due to the COVID-19 pandemic among Hong Kong Chinese older adults is in urgent need.

In light of the changes in health behaviour during COVID-19, it is reasonable to expect that older adults will be less likely to engage in active behaviour due to the decrease in exercise self-efficacy and resilience after traumatic events [11,12]. Exercise self-efficacy refers to individuals' efficacy and confidence in adopting and maintaining continuous exercise behaviour [13] which shows the perception of capabilities of the individual towards exercise participation [13,14] It is documented that older adults' exercise self-efficacy tends to be affected by long-term traumatic injuries, either due to physical (i.e. spinal cord injuries) or sports injuries, which is significantly related to exercise behaviour engagement [1,15]. Nevertheless, a person's exercise self-efficacy was also demonstrated to be affected by affective experiences, behaviour and societal aspects [14]. Based on the above literature, it is probable that the lockdown measures during the COVID-19 pandemic may result in reduced physical activity levels, as well as less social integration and connection among older adults, which could impact their exercise self-efficacy upon returning to the active lifestyle.

According to Sloan's [16] dominance of adults' healthy lifestyle index, healthy lifestyle behaviours involve Physical Activity, Diet, Alcohol Consumption, Smoking, Sleeping Habits, Social Networking, Depression and Stress. Yet, considering to emphasise older adults' perceived efficacy and resiliency towards their health after COVID, participants' general mental

status was not identified in the research outcomes. Additionally, tele-surveys and home-based programs aim to reach more older adults who stay alone, rather than those who are typically active in elderly service centres. It is documented that stay-alone older adults constitute 13.1% of the overall population of older person [15]. Local service centres rarely reach older adults who stay alone and hidden. They were given less care and attention. Therefore, the survey and program are also expected to provide them with a channel to reach out to the community and develop a new active social network.

## Research gap and objectives

With the aforementioned research evidence of previous studies, we conducted a pilot study to first understand the changes of physical activity participation, lifestyle, and overall well-being before, during and after the outbreak of COVID-19 pandemic among older adults, particularly including those stay-alone and hidden older adults. By knowing these facts and findings, the study aims to first, raise the awareness of older adults on reduction of exercise self-efficacy, second, improve their resilience, third, reengage their active behaviours in post-COVID-19 pandemic through an internet home-based exercises program and gradually rebuild new social networks through the internet and face-to-face interaction when the pandemic situation returns to normal. Therefore, the current study is aiming to identify the changes in health behaviours as a result of the COVID-19 pandemic among Hong Kong older adults. This survey could provide an overview picture of the impacts of COVID-19 on older adults' changes in physical activity participation, lifestyle, and overall well-being, particularly for those who live alone and hidden older adults. Most importantly, to identify the older adults' perceived efficacy and resiliency towards their health. This survey could provide evidence-based findings for proposing an internet home-based older adults exercise program in the future.

## Methods

### Participants

In this study, a random tele-survey was conducted to reach out to older adults from May to September 2021 during the COVID-19 social restriction. Based on the population (1,163,153) aged 65 or above in Hong Kong (C&SD, HKSAR government, 2016), the sample size was around 600, with the elderly population proportion at 15.9%, confidence interval of 3 and 95% confidence level. The participants were also recruited by age distribution, with 53% aged "65–74", 17.7% aged "75–79" and 29.3% aged "80 or above.

### Measures

The telephone survey included questions regarding the change in their overall living habit. The content of the question includes two parts. The first part was about age, gender, and living conditions. The second part was about their perceived efficacy toward their health behaviour changes during COVID-19, including eating habits, physical activity participation, sleeping quality, social interaction, bodily pain, self-efficacy, resilience, and open-end questions about the perception of online physical activity programs; and the Eating habits- three items were selected from [17], the survey on people's knowledge of sustainable food and diet. Questions were about the number of meals (including snacks) consumed per week before and during COVID-19; the number of foods intake before three hours of sleep per week before and during COVID-19; and the number of diets out or take-aways per week before and during COVID-19.

   Physical activity- the study modified the International Physical Activity Questionnaire (IPAQ-C) [18] to measure older adults' physical activity level change during the pandemic.

The scale was acceptably reliable (intraclass correlation coefficient of 0.79 and % percentage of the mean score of 26%) [18]. Five items were selected including whether they did sedentary behavior, outdoor exercises, regular physical or leisure activity, housework, and salaries before and during the pandemic. If they said "yes", open-ended questions were asked about what kind of exercises they did.

Sleep- Pittsburgh Sleep Quality Index [19] was used to measure participants' sleep quality, was test consistently correlated with the sleep quality ($\chi2/df > 5$, THE ROOT MEAN SQUARE ERROR OF APPROXIMATION (RMSEA) $> 0.80$) [20]. The selected items included their sleep hour before and during COVID-19; factors that interfere with sleep quality before and during COVID-19; and their evaluation of sleep quality before and during COVID-19.

Social interaction - Social interaction status measurement [21] was used to access older adults' social participation and social contact. This questionnaire was translated to Chinese by [22] and tested with good validity with Cronbach alpha at .605. In this study, we selected four items about the participation in social services, religious activity, and leisure activity from the social participation level before and during COVID-19, to identify the frequency of social contact with friends prior to and during COVID-19.

Bodily pain- we modified the sub-scale of body pain of The Chinese Short From -36 (C-SF-36) and added open-end questions for participants to point out the specific pain of the body and how these pains affect their daily activity before and during COVID-19. For example, "Please specify the main part of your body that is suffering pain"; "How many days have you suffered from chronic physical pain?"; "What is the pain intensity?" From 1 to 10"; and "To what extent are your daily activities affected by those physical pains? From 1 to 10".

Self-efficacy- Five items from Exercises Self-Efficacy Chinese Version [23], with good validity of construct (Cronbach alpha at 0.7) [24] were adopted to measure older adults' exercise self-efficacy change during COVID-19. The items included: their perceptions of having the confidence to attain their physical activity and exercise goals; being physically active when tired; exercising without social support; and motivating themselves to begin physical activity again after a long time. The scale is rated on a 4-point Likert scale from 1 (highly disagree) to 4 (highly agree).

Resilience- The Chinese version of the Resilience Scale (CRS) was adopted to measure the level of resilience among Hong Kong older adults, with acceptable content validity (I-CVI $> 0.83$) [25]. This valid scale was also used for measuring Chinese older adults' resilience by scholars [26]. We modified the questionnaire into four items, such as "I believe I have the ability to overcome difficulties"; "I believe I am physically able to perform outdoor physical activities"; "My family provides me with a lot of support when things get tough"; "I am open to and accept the unfortunate things in life". The scale was rated on a 4-point Likert scale from 1 (highly disagree) to 4 (highly agree).

## Procedure

Before collecting data, ethics had been approved by Research Ethics Committee. All telephone surveys were conducted by mobile phone interviewers. In addition to the on-site supervision and random checking, voice recording, screen capturing, and camera surveillance was used to monitor interviewers' performance and data quality. Cell phone numbers were generated using known prefixes assigned to telecommunication service providers under the Numbering Plan provided by the Office of the Communication Authority (OFCA). The randomization was also stratified by age and sex, according to the 2016 Population By-census [27]. Three to five mobile phone calls were made to non-responders. Invalid numbers were eliminated according to digital and manual dialling records to produce the final sample.

## Data analysis

Descriptive statistics, such as frequency, mean and standard deviation, as well as the paired sample t-test were analysed using SPSS to discover the older adults' health behaviour change before and during COVID-19. For the open-ended questions, trained research assistants developed coding frameworks; they categorized and labelled the participants' opinions regarding their perception of future online exercises program about content and elements.

Moreover, the exploratory network analysis using the Gaussian Graphical Model was conducted to explore the relationship between the healthy lifestyle variables before and during in R, version 4.0 [28]. For missing data, multiple imputation (using the "mice" package) was used to substitute the missing data with the mean score, implying that all variables were continuous. The network was estimated and investigated using the "bootnet" packages[29]. In order to maximize the regularization of the parameters, the EBICglasso was used to fit a Gaussian graphical model with the extended Bayesian information criteria. Boostrapping (1000 bootstraps) was used to assess the network's accurateness and robustness (stability). The networks are visualized using the qgraph package, in order to reveal the centrality, strength, as well as the stability of the relationships. The weight of the edges between the variables were also be indicated, such that the level of relationships between different variables, as well as between before and during COVID-19, can be revealed. In the graph, blue edges represented positive correlations while red edges represent negative correlations. Additionally, the ggmFit function was applied to reveal the model fit of the estimated network. The estimated network was transformed into a lavaan model to investigate the possible causal paths between the variables.

## Results

### Demographic information

A total of 664 participants were involved in the telephone survey. There were 213 males and 451 females, of which mean age at 74.37. The mean score of healthy lifestyle behaviour variables was displayed in the demographic information (Table 1). Participants showed a reduction in exercise self-efficacy (t(438) = 6.03, $p<0.001$), leisure activities (t(663) = 7.67, $p<0.001$), unhealthy eating habits (t(134 = 4.09. $p<0.001$) and social network (t(663) = 16.65, $p<0.001$), which was well expected. Participants also showed a higher rate in feeling bodily pain during COVID-19, yet not statistically significant. Participants indicated that they slept more during the COVID-19 pandemic (t(584) = -11.03, $p<0.001$), thus resulting in a higher level of sleeping quality (t(663) = -4.05, $p<0.001$).

### The estimated network

With edge correlation stability coefficient at 0.75 and node strength correlation stability coefficient at 0.36, it indicated that the network will be remained stable despite a 75% change in the dataset. All variables of before and during COVID-19 are significantly connected (Fig 1). Among the nodes, social network before COVID-19 (strength coefficient [SC] = 1.47), resilience before COVID-19 (strength coefficient [SC] = 1.10), and exercise efficacy during COVID-19 (strength coefficient [SC] = 1.08). Social networking before COVID-19 had the highest centrality degree (1.28), closeness (0.0023) and betweenness (88), which indicated the possible impact of social networking on participants' healthy lifestyle behaviours, either before or during COVID-19. Centrality plot, centrality stability plot, and edge stability are presented in the supporting information (S1-S3 Figs in S1 File). The centrality measures for all variables, and the weight matrix of all edges can be found in the supporting information (S1 and S2 Tables in S1 File). Among the edges in the network (excluding the relationships between

**Table 1. Demographic information and the summary of the healthy lifestyle behaviours variables.**

| | | Frequency | Mean | SD |
|---|---|---|---|---|
| Gender | Female | 451 | / | / |
| | Male | 213 | / | / |
| Age | 65–74 | 364 (54.8%) | / | / |
| | 75–79 | 111(16.7%) | / | / |
| | 80 or above | 189(28.5%) | / | / |
| | | | 74.37 | 7.02 |
| Sleeping Time | Before | / | 393.09 | 99.81 |
| | During | / | 398.27 | 98.74 |
| Sleeping Quality | Before | / | 2.2500 | .70 |
| | During | / | 2.3133 | .74 |
| Eating Habits | Before | / | 2.62 | 0.69 |
| | During | / | 2.51 | 0.69 |
| Leisure Activities | Before | / | 1.59 | 0.20 |
| | During | / | 1.55 | 0.18 |
| Social Network | Before | / | 1.68 | 0.81 |
| | During | / | 1.28 | 0.51 |
| Bodily Pain | Before | / | 5.60 | 2.42 |
| | During | / | 5.63 | 2.42 |
| Exercise Efficacy | Before | / | 3.11 | .68 |
| | During | / | 2.99 | .75 |
| Resilience | Before | / | 3.19 | .52 |
| | During | / | 3.19 | .52 |

before and during COVID-19), the strongest edge in the network was social networking and leisure activity engagement before and during COVID-19 (edge-weight = 0.25; 0.10). The next strongest edge was exercise efficacy and resilience before and during COVID-19 (edge-weight = 0.17;0.14). Furthermore, the estimated network of healthy lifestyle behaviours before and during the COVID-19 showed excellent model fit, with $X^2(142.82/120) = 1.19$, comparative fit index (CFI) = 0.99, THE TUCKER-LEWIS INDEX (TLI) = 0.97, THE ROOT MEAN SQUARE ERROR OF APPROXIMATION (RMSEA) = 0.062, SRMR = 0.008.

## The causal paths

The healthy lifestyle behaviour before and during the COVID-19 network was transformed into a lavaan model, in order to investigate the potential causal relationship between variables. The transformed network showed an adequate model goodness-of-fit, with COMPARATIVE FIT INDEX (CFI) = 0.93, THE TUCKER-LEWIS INDEX (TLI) = 0.89, THE ROOT MEAN SQUARE ERROR OF APPROXIMATION (RMSEA) = 0.19, SRMR = 0.056. Within the network, meaningful paths were identified. Firstly, the level of resilience before COVID-19 was significantly influenced by bodily pain directly ($\beta = 0.39$, $p<0.001$), and sleeping quality indirectly through level of leisure activities participation ($\beta = 0.18$, $p<0.001$). Secondly, level of resilience during COVID-19 was seen as significantly influenced by healthy lifestyle behaviours, including directly by eating habits ($\beta = 0.6$, $p<0.001$), and indirectly by social networking through eating habits ($\beta = 0.004$, $p<0.001$). Additionally, level of resilience in both before and during COVID-19 had an impact on older adults' social networking and leisure activities participation; people with higher levels of resilience tended to participate in more social networking ($\beta = 0.37–38$) and leisure activities ($\beta = 0.31$, $p<0.001$). Following the effect towards

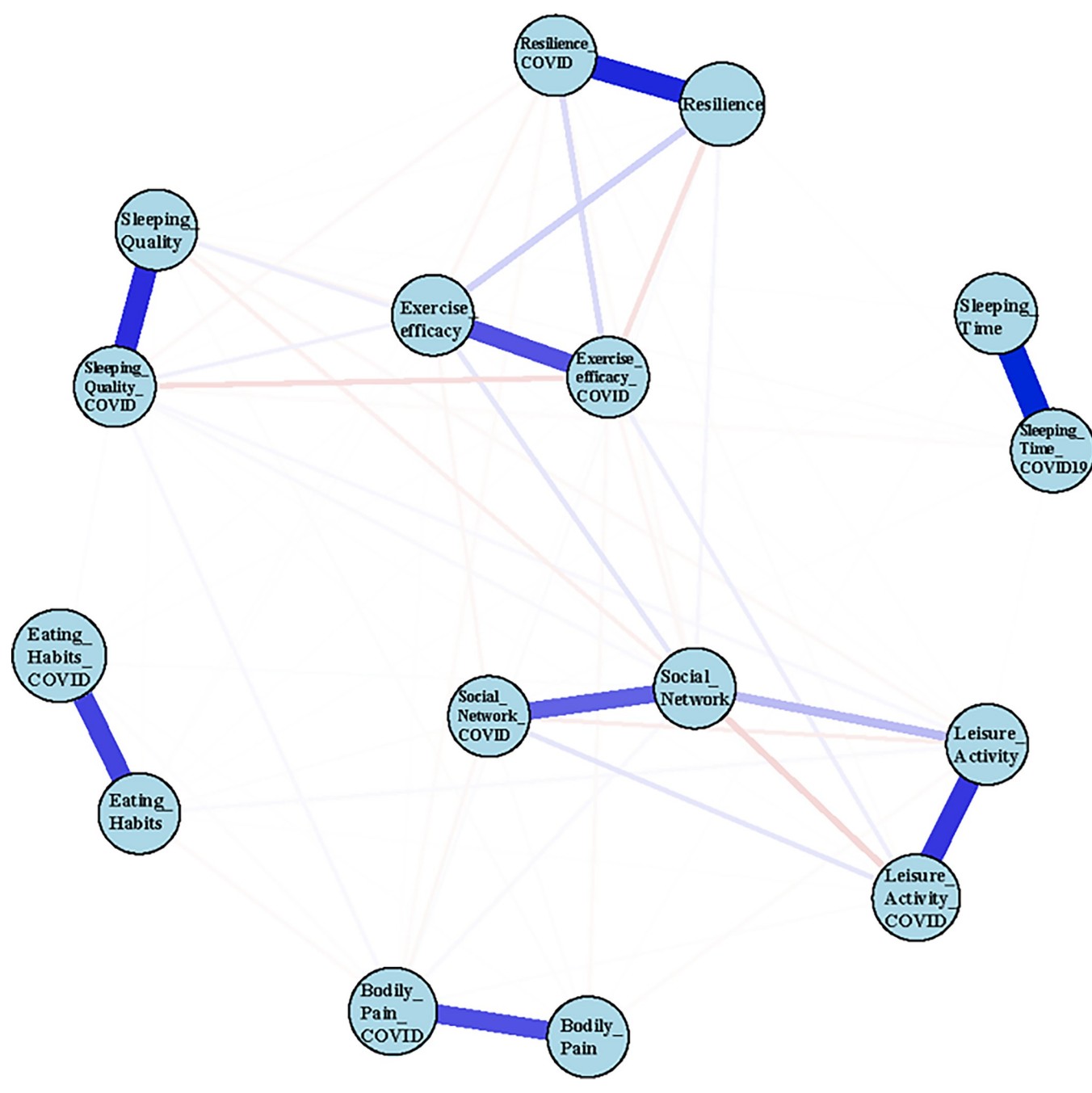

**Fig 1. The estimated network.**

social networking led by resilience, social network showed a positive influence on older adults' sleeping quality in both before and during the COVID-19 (β = 0.67, $p<0.001$; β = 0.63, $p<0.001$). Thirdly, exercise efficacy was both determined by bodily pain before and during COVID-19, one negatively (before: β = 0.61, $p<0.001$) and one positively (during: β = 0.64, $p<0.001$). Finally, older adults' sleeping quality would positively affect their eating habits both before (β = 0.49, $p<0.001$) and during COVID-19 (β = 0.20, $p<0.001$). Supporting information (S3 Table in S1 File) displays the summary of the connected paths.

## Discussion

The current study investigated the healthy lifestyle behavioural changes of Hong Kong older adults before and during COVID-19. The use of telephone surveys benefits the study with a relatively large sample size thus being representative. Hong Kong older adults showed a significant decline in exercise self-efficacy, leisure activities participation (physical activity, walking, volunteering) and social networking during COVID-19 as compared with the time prior. Hong Kong older adults also showed reduced unhealthy eating habits during COVID-19, such as eating out less. Hong Kong older adults slept more during COVID-19, which resulted in a higher level of sleeping quality. However, they found no significant differences in level of resilience. In the estimated network, healthy lifestyle behaviours were found to be significantly associated with each other, especially the association between leisure activities and social networking, as well as exercise self-efficacy and resilience. Furthermore, the lavaan model has also displayed various meaningful pathways that are worth discussing.

In the existing literature, the COVID-19 pandemic has substantially affected older adults' lifestyle behaviours, including physical activity engagement, eating habits and social engagement [30–33]. Most of the literature claimed the effect of COVID-19 on older adults' physical inactivity [33,34] as well as less social engagement [30,31,34] which were aligned with the current study. However, the results of the changes in eating habits and sleeping quality were varying. A narrative review has shown that older adults tended to have a poor sleeping quality during COVID-19 [34], yet the older adults of Hong Kong showed a better sleeping quality during COVID-19 with the social distancing measure. Moreover, the eating habits changes outcomes were inconsistent.[35,34,33] Research indicated that older adults reported more snacks and undernutrition meals intake during COVID-19 [35]; however, other recent research studies revealed that older adults showed stress towards health-related issues thus resulting in healthier eating [36]. People, in general, showed positive dietary changes and showed higher motives towards food choices [32] due to the COVID-19 pandemic. Research studies have come up with the same conclusion despite the variations, which is that these changes have contributed to a higher level of stress among older adults and that physical activity interventions, free-play physical activity interventions, and relaxation techniques should be encouraged [37,38].

In the estimated network, the reasons behind the strong association between social networking and leisure activities were obvious. Leisure activities that have been mentioned in the telephone survey include physical activity, walking and volunteer work were also seen as the common social activities of Hong Kong older adults. While, the association between exercise self-efficacy and resilience among older adults were less likely to be mentioned in the existing literature. In fact, self-efficacy is shown as a protective predictor of older adults' resilience [39]. Research on chronic pain patient showed that people with higher level of pain self-efficacy contributes to their psychological resilience in overcoming the pain intensity, as well as treatment [40]. Hence, it can assume that older adults with a higher level of exercise self-efficacy is associated with higher level of resilience in overcoming one's health-related issue, including bodily pain, or even recovering from COVID-19.

Resilience, the ability to overcome and recover from stress and obstacles coming from physical and mental loss, as well as the mentality of confronting mortality, has long been regarded as an important component of the aging process. Previous studies have indicated the strong association between functionality of healthy lifestyle behaviours and resilience [41,42]. Similarly, the current study has discovered significant pathways between lifestyle variables and resilience among Hong Kong older adults, both before and during COVID-19. The current research indicates that the level of bodily pain would influence the level of resilience [43,44].

Individuals who are better at adapting to pain have a higher level of psychological resilience, and self-efficacy has played a function as a mediator in this relationship [44]. Other than bodily pain, resilience was shown to be influenced by different lifestyle behaviours, including eating habits, sleeping quality and social networking. Existing research has shown that adherence to appropriate eating patterns leads to increased physical resilience [45], as well as that a higher BMI predicts a poorer level of resilience among healthy older individuals [45,46]. Despite the fact that research on the relationship between social networking and resilience has tended to be related to trauma, disasters, or migration, there is evidence that sleep disturbance is associated with lower levels of perceived social support and resilience among general adults [47]. There was well-documented literature that mirrored the findings of the current study in terms of the impact on eating behaviours. Research indicated that older adults who consumed more fruits and vegetables, as well as following the Mediterranean diet with better nutrients, reported better sleep quality [48,49]. In addition, there are noticeable changes in people's eating habits and sleeping patterns during COVID-19 [50].

An interesting pathway was demonstrated in the current study, which is the relationship between bodily pain and exercise self-efficacy before and during COVID-19. It is not surprising that bodily pain has a negative impact on older adults' exercise self-efficacy most of the time. Improved bodily pain is expected to improve individual exercise self-efficacy [51], which individuals tended to have a higher confidence in their exercise and physical ability with less bodily pain. Exercise interventions were also seen as a significant tool in relieving physiological pain thus enhancing exercise self-efficacy [52]. During the COVID-19 pandemic, however, older persons demonstrated that having higher physiological pain had a positive impact on exercise self-efficacy, which existing literature could hardly explain. It can be interpreted as COVID-19 has increased older adults' perceived stress towards health, indicating that exercise self-efficacy might be maintained or even enhanced in spite of their body pain.

This study provided implications for the healthy lifestyles of older adults. Firstly, the community needs to provide more healthy activities for older adults to maintain their healthy lifestyle, as unexpected disasters cannot be accurately predicted and educating older adults on coping with health risk events is necessary, for example, increasing resilience for positive coping, technical assistance and targeted government and community support may protect older adults from suffering during a pandemic [53]. Second, some older patients may experience post-COVID-19 conditions, a guide for restoring movement should be designed for older adults, for example, cooperating with clinics and hospital, person-tailor comprehensive recovery plan could be designed from the beginning phase, building phase and being phase [54].

To conclude, the current study has provided an overview of healthy lifestyle behaviour changes of older adults during the COVID-19 pandemic, using the exploratory network analysis. Yet, the limitations of the study should be acknowledged. The current study is conducted through a telephone survey among older adults. In order to reduce response fatigue and dropout, it did not apply the full scale to some of the variables, such as resilience, exercise self-efficacy, and social networking. Furthermore, the telephone survey was conducted during the social distancing measures, which may result in a certain amount of recall bias among respondents when recalling healthy lifestyle behaviours before COVID-19. Concerning the unexpected outcomes from the relationship between bodily pain and exercise self-efficacy, one potential limitation is the inability to distinguish between injuries and muscle pain after exercise. Moreover, although the present study is not regarded as novel internationally, it may be representative of the Hong Kong community. As a result, based on the outcomes of the study, the effect of COVID-19 on older adults' lifestyle changes was identified. Home-based healthy lifestyle interventions, such as home-based physical activity plus psychoeducation workshop, online physical or mental therapeutic consultation services, stress management interventions

or other user-friendly online psychosocial platform, should be launched to facilitate the needs of older adults. There is an increasing acceptance by older adults of changing from face-to-face to virtual interactions; therefore, the electronic platform for healthy living should also include older adults in order to broaden the scope of active ageing during the pandemic.

## Supporting information

**S1 File.**
(ZIP)

## Author Contributions

**Conceptualization:** Ming Yu Claudia Wong.

**Data curation:** Ka Man Leung.

**Methodology:** Kai-ling Ou.

**Software:** Ming Yu Claudia Wong.

**Writing – original draft:** Ming Yu Claudia Wong, Kai-ling Ou.

**Writing – review & editing:** Ming Yu Claudia Wong.

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
