## [Decision Letter · Decision Letter 0]

2 Oct 2022

PONE-D-22-18963Investigating and Promoting Health Behaviors Reactivity Among Hong Kong Older Adults in the Post-COVID-19 Era: An Exploratory Network AnalysisPLOS ONE

Dear Dr. Ou,

Thank you for submitting your manuscript to PLOS ONE. After careful consideration, we feel that it has merit but does not fully meet PLOS ONE’s publication criteria as it currently stands. Therefore, we invite you to submit a revised version of the manuscript that addresses the points raised during the review process.

We look forward to receiving your revised manuscript.

Kind regards,

Tai Ming Wut

Academic Editor

PLOS ONE

Journal Requirements:

“Funding sources and related paper presentations:

Hong Kong Baptist University: Research development fund

RC-RDF-RNHA202103 ~ Research Network on Healthy Ageing - Chung Pak Kwong”

“NO authors have competing interests

Enter: The authors have declared that no competing interests exist.”

“Thanks to The Centre for the Advancement of Social Sciences Research (CARS) of Hong Kong Baptist University for providing professional technical support in conducting and supervising mobile phone interviews.

Sponsor's Role

This research is sponsored by Faculty Research Development Fund (Research Network on Healthy Ageing)   (HK$135,000), Hong Kong Baptist University. The sponsor provided advice on the direction of our initial research and the fund supported the cost of the telephone  survey”

“Funding sources and related paper presentations:

Hong Kong Baptist University: Research development fund

RC-RDF-RNHA202103 ~ Research Network on Healthy Ageing - Chung Pak Kwong”

Reviewers' comments:

Reviewer's Responses to Questions

**Comments to the Author**

1. Is the manuscript technically sound, and do the data support the conclusions?

Reviewer #1: Yes

Reviewer #2: Yes

2. Has the statistical analysis been performed appropriately and rigorously? 

Reviewer #1: Yes

Reviewer #2: Yes

3. Have the authors made all data underlying the findings in their manuscript fully available?

Reviewer #1: Yes

Reviewer #2: Yes

4. Is the manuscript presented in an intelligible fashion and written in standard English?

Reviewer #1: Yes

Reviewer #2: Yes

5. Review Comments to the Author

Reviewer #1: The methods and results should be SIMPLE PAST not past perfect or any other tense.

More citations and evidence are needed to support reliability, validity, psychometric properties, and factor structure of the measures and questionnaires.

Too many abbreviations. Define them, and reduce them.

Fit data of various models to be presented in a table.

What are * when p value is reported in the text of the results? Is it a typo?

The results may not hold for older old (80 yrs), low SES and high SES, and ethnic groups. Please test invariance by these factors.

Reviewer #2: Revisions are required. Please address the following comments and questions.

1. Strengthen theoretical background. Elaborate in more depth. Connect theory with research design and discussion.

2. Literature review is on the brief side. Requires more discussion, analysis, and citations.

3. List the objectives in a separate section;

4. Include a separate research gap section;

5. List the hypotheses;

6. Line 128: Would COVID-19 be considered as a traumatic event?

7. Provide reliability and validity figures for the psychometric instruments;

8. Add response rate; “has proved”

9. Line 328: tone down

10. Line 342: add citations

11. Discussion, implications, and future research directions sections: Elaborate in more depth.

12. Lines 47 and 399: tone down “representative of the Hong Kong community.”

13. Line 35: “tele survey” -> telephone survey

14. “sleeping quality” -> sleep quality

6. PLOS authors have the option to publish the peer review history of their article (what does this mean?). If published, this will include your full peer review and any attached files.

Reviewer #1: No

Reviewer #2: No

---

## [Author Response · Author response to Decision Letter 0]

28 Mar 2023

Reply to reviewer

Reviewer #1: The methods and results should be SIMPLE PAST not past perfect or any other tense.

Thanks for your comments, the tense has been revised. 

More citations and evidence are needed to support reliability, validity, psychometric properties, and factor structure of the measures and questionnaires.

Supported references have been added for the corresponding methods. 

Too many abbreviations. Define them, and reduce them.

Those abbreviations were defined.

Fit data of various models to be presented in a table.

The authors do understand the reviewer’s concern, however, as there’s only one model to be shown, therefore, it was directly shown inside the manuscript. 

What are * when p value is reported in the text of the results? Is it a typo?

The Typo has been revised. 

The results may not hold for older old (80 yrs), low SES and high SES, and ethnic groups. Please test invariance by these factors.

The authors agreed with the reviewer’s concern. While, the study has included around 30% of older adults aged 80 or above, which is much higher than that of aged between 75-79, hence we believe the results could hold for all older adults age groups. In spite of that, the invariance model between age groups have been tried, yet the observations of each group is too small with lots of paths and indicators to perform the model invariance test. 

Unfortunately, due to the focus on COVID-19 and the majority low monthly income of older adults, the social status and ethnicity of Hong Kong older adults were not involved in the telephone survey. 

Reviewer #2: Revisions are required. Please address the following comments and questions.

1. Strengthen theoretical background. Elaborate in more depth. Connect theory with research design and discussion.

2. Literature review is on the brief side. Requires more discussion, analysis, and citations.

The authors well understood the suggestion offered by the reviewer. The authors have added more literature and citation regarding the importance and phenomenon of Chinese older adults’ physical activity and other healthy lifestyle changes and effect during COVID-19. However, the current manuscript is being treated as a deductive explorative means to investigate the relationship between the healthy lifestyle indicators in a data-driven approach, yet not theory-driven. The social networking method is considered as a tool to demonstrate the relationships between the indicators but not aiming to present it as a theory or conceptual framework. Therefore, the authors have managed to add the definition of the healthy lifestyle behavior, and the importance of older adults’ perceived exercise efficacy and resilience. 

3. List the objectives in a separate section;

4. Include a separate research gap section;

The research objectives and research gap were separated as suggested. 

5. List the hypotheses;

Hypotheses are not applicable to explorative social network analysis with the research outcome as data-driven; and with no research in regards to COVID-19 has been done on Hong Kong older adults’ perceived exercise efficacy and resilience. 

6. Line 128: Would COVID-19 be considered as a traumatic event?

Thanks for your concerns. Similar to previous pandemics, such as SARS, COVID is a large-scale and highly contagious virus, and the city experienced a lockdown during the epidemic. some literature also highlights that it is responsible for some post-traumatic symptoms and some emotional reactions such as anger, anxiety, and helplessness. Hong Kong has experienced five waves of the epidemic and many elderly people have died as a result, so we believe that COVID is considered to be traumatic to some extent. (Bridgland et al., 2021). 

7. Provide reliability and validity figures for the psychometric instruments;

Some indexes for representing reliability and validity were added. 

8. Add response rate; “has proved”

9. Line 328: tone down

The new sentence: “Most of the literature claimed the effect of COVID-19 on older adults’ physical inactivity”

10. Line 342: add citations

Reference had been added.

11. Discussion, implications, and future research directions sections: Elaborate in more depth.

Please see the implication and future direction added in the discussion. 

12. Lines 47 and 399: tone down “representative of the Hong Kong community.”

The new sentence: “it may be representative of the Hong Kong community”

13. Line 35: “tele survey” -> telephone survey

Revised

14. “sleeping quality” -> sleep quality

Revised

Reference

Bridgland, V. M., Moeck, E. K., Green, D. M., Swain, T. L., Nayda, D. M., Matson, L. A., ... & Takarangi, M. K. (2021). Why the COVID-19 pandemic is a traumatic stressor. PloS one, 16(1), e0240146.

---

## [Decision Letter · Decision Letter 1]

16 Oct 2023

Investigating and Promoting Health Behaviors Reactivity Among Hong Kong Older Adults in the Post-COVID-19 Era: An Exploratory Network Analysis

PONE-D-22-18963R1

Dear Kai-ling Ou,

We’re pleased to inform you that your manuscript has been judged scientifically suitable for publication and will be formally accepted for publication once it meets all outstanding technical requirements.

Kind regards,

Tai Ming Wut

Academic Editor

PLOS ONE

Additional Editor Comments (optional):

Reviewers' comments:

Reviewer's Responses to Questions

**Comments to the Author**

1. If the authors have adequately addressed your comments raised in a previous round of review and you feel that this manuscript is now acceptable for publication, you may indicate that here to bypass the “Comments to the Author” section, enter your conflict of interest statement in the “Confidential to Editor” section, and submit your "Accept" recommendation.

Reviewer #2: All comments have been addressed

2. Is the manuscript technically sound, and do the data support the conclusions?

Reviewer #2: Yes

3. Has the statistical analysis been performed appropriately and rigorously? 

Reviewer #2: Yes

4. Have the authors made all data underlying the findings in their manuscript fully available?

Reviewer #2: Yes

5. Is the manuscript presented in an intelligible fashion and written in standard English?

Reviewer #2: Yes

6. Review Comments to the Author

Reviewer #2: (No Response)

7. PLOS authors have the option to publish the peer review history of their article (what does this mean?). If published, this will include your full peer review and any attached files.

Reviewer #2: No

---

## [Editor Report · Acceptance letter]

24 Oct 2023

PONE-D-22-18963R1 

Investigating and Promoting Health Behaviors Reactivity Among Hong Kong Older Adults in the Post-COVID-19 Era: An Exploratory Network Analysis 

Dear Dr. Ou:

I'm pleased to inform you that your manuscript has been deemed suitable for publication in PLOS ONE. Congratulations! Your manuscript is now with our production department. 

Kind regards, 

on behalf of

Dr. Tai Ming Wut 

Academic Editor

PLOS ONE